# Structural mechanism of signal transduction in a phytochrome histidine kinase

Weixiao Yuan Wahlgren[1], Elin Claesson[1], Iida Tuure [2], Sergio Trillo-Muyo [3], Szabolcs Bódizs [1], Janne A. Ihalainen [2], Heikki Takala [2,4] ✉ & Sebastian Westenhoff [1,5] ✉

Phytochrome proteins detect red/far-red light to guide the growth, motion, development and reproduction in plants, fungi, and bacteria. Bacterial phytochromes commonly function as an entrance signal in two-component sensory systems. Despite the availability of three-dimensional structures of phytochromes and other two-component proteins, the conformational changes, which lead to activation of the protein, are not understood. We reveal cryo electron microscopy structures of the complete phytochrome from *Deinoccocus radiodurans* in its resting and photoactivated states at 3.6 Å and 3.5 Å resolution, respectively. Upon photoactivation, the photosensory core module hardly changes its tertiary domain arrangement, but the connector helices between the photosensory and the histidine kinase modules open up like a zipper, causing asymmetry and disorder in the effector domains. The structures provide a framework for atom-scale understanding of signaling in phytochromes, visualize allosteric communication over several nanometers, and suggest that disorder in the dimeric arrangement of the effector domains is important for phosphatase activity in a two-component system. The results have implications for the development of optogenetic applications.

Light is essential for life and all living organisms have developed intricate systems to detect and adapt to it. At the molecular level, this is achieved by photoreceptor proteins. Phytochromes are a protein superfamily in bacteria, plants, and fungi, and function by photo-switching between states that absorb red light (Pr) and far-red light (Pfr)[1–6]. These states have differential signaling activity. Conserved over the entire superfamily and exemplified by the phytochrome from *D. radiodurans*, phytochromes share a photosensory core module, consisting of PAS, GAF, and PHY domains and a bilin chromophore (Fig. 1), which is connected to a variable effector module via helical linkers[3]. The structure of the photosensory core is very similar in plant and bacterial phytochromes[7–13] but diverges in the context of full-length proteins[14]. Phytochromes are highly interesting targets for gene optimization of plants, are used as near-infrared fluorescent

markers in microscopy[15], and have been utilized in optogenetic applications[16,17].

Many bacteriophytochromes are histidine kinases or phosphatases in two-component systems (TCSs)[18,19]. TCSs sense environmental cues in all kingdoms of life. They are particularly important in microorganisms and plants, where they provide for cell growth, survival, and pathogenicity, and are implicated in multidrug and antibiotic resistance[20]. TCSs consist of sensor kinases, which send phospho-signals to so-called response regulators. A highly pertinent question is how the kinases achieve this signal transduction at the atomic level. Crystal structures of proteins involved in TCS have resulted in a number of partially conflicting conformational mechanisms for signal transduction[21–26], one of which is that the region around the active site for phosphate transfer is destabilized[27]. Opposing the notion that one

[1]Department of Chemistry and Molecular Biology, University of Gothenburg, Gothenburg, Sweden. [2]Department of Biological and Environmental Science, Nanoscience Center, University of Jyvaskyla, Jyvaskyla, Finland. [3]Department of Medical Biochemistry and Cell Biology, University of Gothenburg, Gothenburg, Sweden. [4]Faculty of Medicine, Anatomy, University of Helsinki, Helsinki, Finland. [5]Department of Chemistry—BMC, Biochemistry, Uppsala University, Uppsala, Sweden. ✉e-mail: heikki.p.takala@jyu.fi; westenho@chem.gu.se

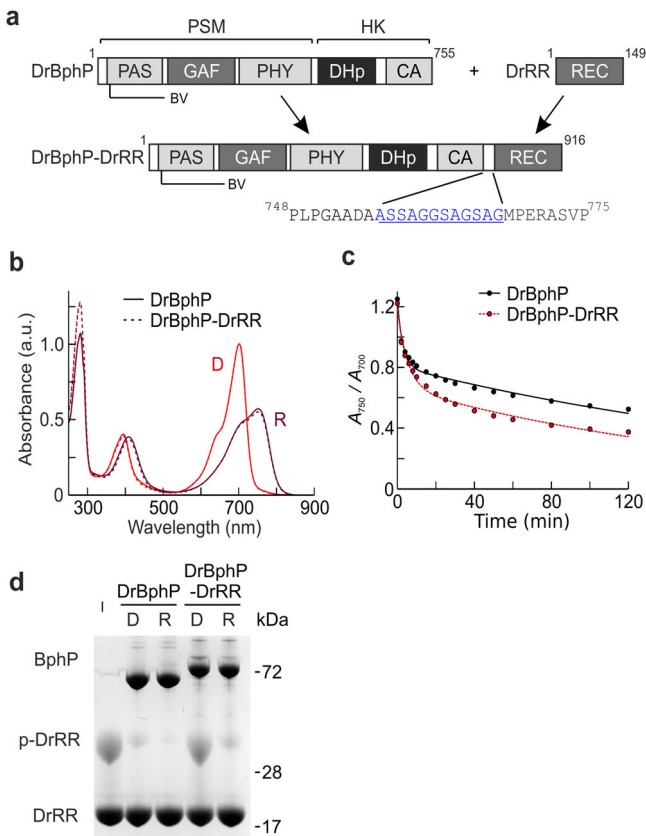

**Fig. 1 | Spectra and function of *Dr*BphP and *Dr*BphP-*Dr*RR. a** The domain organization and the linker configuration are shown with the position of the biliverdin (BV) chromophore indicated. CA catalytic ATP-binding, DHp dimerization and histidine phosphotransfer, GAF cGMP phosphodiesterase-adenylate cyclase FhlA, HK histidine kinase, PAS Per-ARNT-Sim, PHY phytochrome-specific, PSM photosensory module, REC receiver domain, RR response regulator. **b** Absorption spectra of both proteins recorded in dark (D) or after saturating 655 nm red light (R). The spectra are normalized at the peak at 700 nm of the corresponding dark spectrum. Source data are provided as a Source Data file. **c** Dark reversion of *Dr*BphP and *Dr*BphP-*Dr*RR. Response regulator fusion increases the *Dr*BphP dark reversion, which resembles the addition of soluble *Dr*RR to *Dr*BphP (see Supplementary Fig. 1b)[35]. Source data are provided as a Source Data file. **d** Phos-tag assay of phosphatase activity[35]. In the assay, the retention of phosphorylated protein is slower than its non-phosphorylated counterparts. Here, all samples were supplemented with the same amount of phospho-*Dr*RR (p-*Dr*RR) and the change in its final amount implies either kinase (increase) or phosphatase (decrease) activity. The gel shows that *Dr*BphP-*Dr*RR dephosphorylates p-*Dr*RR in its red light-illuminated state (lane R) and the activity is stalled in the darkness (lane D). The phosphatase activity of the fusion protein resembles that of *Dr*BphP, although is generally weaker because of competition between the fused and free *Dr*RR. See Supplementary Fig. 1c for an extended gel. Source data are provided as a Source Data file.

clearly defined cascade of structural changes leads to signaling, it has been proposed that allosteric coupling between the different modules controls the activity of the effector domains[25,28], but a structural understanding of this concept is missing.

Reminiscent of the state-of-the-art for TCSs, the conformational changes that guide phytochrome activation are not understood. Structures have been obtained for the photosensory core fragment of bacteriophytochromes in Pr and Pfr states[10–12], pinpointing isomerization of the D-ring of the bilin cofactor and refolding of the so-called PHY tongue, which is a hairpin structure that extends from the PHY domain onto the chromophore region, as outstanding features of the photoconversion[13]. Finer mechanistic insight near the chromophore has been obtained using spectroscopy and crystallography[29–32].

Signal transduction from the photosensory core to the effector domains has been proposed to be guided by a monomerization mechanism[33], or by a shift of register in the dimeric binding interface of the sensor-effector linker[34]. However, solid progress is limited by a lack of structures of full-length phytochromes in active and resting states.

In this work, we use single-particle cryo-electron microscopy (cryo-EM) to solve the structure of the full-length phytochrome from *D. radiodurans* (*Dr*BphP) in the resting (Pr) and photoactivated (Pfr) states. We present models in both states, revealing a mechanism of photoactivation.

## Results

For the single-particle cryo-EM experiment, we used a fusion of *Dr*BphP with its response regulator protein (abbreviated *Dr*BphP-*Dr*RR) via a 12 amino acid tether (Fig. 1a). This gave a better spatial resolution in the reconstructed maps compared to the unfused *Dr*BphP. We assume that *Dr*RR stabilizes the *Dr*BphP structure by transiently interacting with its cognate binding site, but that the structure of *Dr*BphP is otherwise unaffected. Compared to the wild type, the fusion protein has similar absorption spectra (Fig. 1b), a slightly enhanced dark reversion rate (Fig. 1c), and similar but reduced phosphatase activity in Pfr (Fig. 1d)[35].

Figure 2 shows the reconstructed electron density map from 117,297 single-particle images of *Dr*BphP-*Dr*RR in Pr, with grids prepared from monodisperse protein in darkness (Supplementary Fig. 1a). The selection and refinement process is summarized in Supplementary Fig. 2. Incomplete particles were removed based on 2D classifications. Three 3D classes were obtained from ab-initio classification. Classes 2 and 3 contained incomplete densities and were discarded. Homogeneous refinement of class 1 followed by local refinement implemented in cryoSPARC v3.1.0 gave a resolution of 3.6 Å, and local refinement with a mask over the PAS-GAF-PHY-neck region resulted from a resolution of 3.4 Å. Both maps had a local difference in resolution (Supplementary Fig. 6). The clear densities corresponding to PAS-GAF-PHY-DHp, including the "neck"-linker between PHY and DHp (Supplementary Fig. 6), allowed us to build a model with confidence for these domains (Table 1). The modeled structure shows the signatures expected for the Pr state, including the PHY tongue in a β-sheet conformation (Fig. 2b) and the biliverdin in 15*Z* conformation (Fig. 2c and Supplementary Fig. 5c)[12,13]. The DHp domains show a dimeric and symmetric conformation, which is often found in histidine kinases[23]. The two helices of a DHp domain can be connected in two ways in the four helical bundles of the dimer. We chose to link the two DHp helices following the crystal structure of the *Thermotoga maritima* histidine kinase[26], but note that the reconstructed electron densities would also support the alternative linkage of the helices. The structure of the PAS-GAF-PHY domains is almost identical to the crystal structure of the corresponding truncation of *Dr*BphP (Supplementary Fig. 3e)[13,36]. We also find two strong and two weaker patches of densities, which are arranged symmetrically around the DHp domains. They resemble the crystal structure of the *Thermotoga maritima* histidine kinase in complex with its response regulator (pdb entry 3DGE)[36], and we assign the patches to the REC and CA domains (Supplementary Fig. 5a).

We continued by solving the structure of *Dr*BphP-*Dr*RR in its Pfr state. The grids were prepared as for Pr, but under illumination with red light (see Materials and Methods). These conditions yield proteins in a photoequilibrium of Pr/Pfr. The selection and refinement process is summarized in Supplementary Fig. 4. Pfr and Pr conformations were identified and separated as 3D classes in the data (see Supplementary Fig. 4 for details), yielding a reconstructed electron density map for Pfr from 98,050 single-particle images of *Dr*BphP-*Dr*RR (Fig. 3). From the reconstruction we estimated that ~50% of intact proteins on the grid were in Pfr (Supplementary Fig. 4). Although we did not impose any symmetry in the classification, no mixed Pr/Pfr conformations were resolved as judged by the densities at the tongue region. We used local

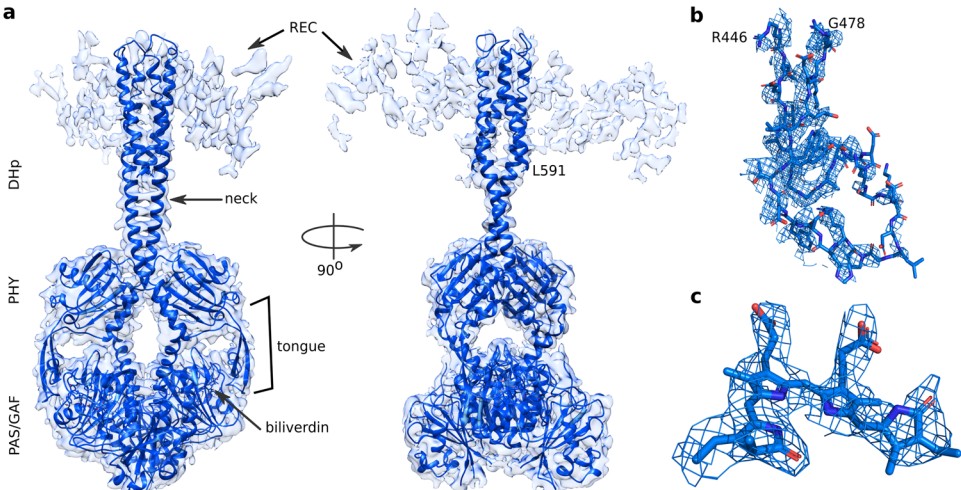

**Fig. 2 | The cryo-EM structure of *Dr*BphP-*Dr*RR in Pr. a** The reconstructed electron density of Pr full-length *Dr*BphP-*Dr*RR is shown with the structural model which was refined up to residue Leu591. The electron density of the CA domains is below the presented contour level. See Supplementary Fig. 5a for CA and REC domain assignment. **b** The PHY tongue has adopted a β-sheet form characteristic of Pr-state phytochromes. **c** The density of the BV chromophore supports a 15*Z*-conformation (compare Supplementary Fig. 5c). The densities of the tongue and BV are from the cryo-EM map with the local refinement of the PSM and neck region (Pr PSM, see Supplementary Fig. 2).

refinement with a mask of the region up to an approximate residue number of 521, which comprises the PAS-GAF-PHY domains and the helical neck to obtain a map with an average resolution of 3.5 Å (Supplementary Fig. 7). In these densities, we built a protein model consisting of PAS-GAF-PHY-neck (residues 22–521) (Fig. 3a, Table 1). The protein is confirmed to be in Pfr as the PHY tongue appears as an α−helix (Fig. 3b)[13] and the chromophore is best modeled in 15*E* conformation (Fig. 3c and Supplementary Fig. 5d)[11]. Unexpectedly, the 3D arrangement of the PAS-GAF-PHY domains is notably different from the crystal structures of the PAS-GAF-PHY truncation in Pfr (Supplementary Fig. 3f)[13]. The electron density in the histidine kinase output domains is asymmetric and weaker compared to the corresponding density in the Pr state. This made an assignment of the output domains challenging and our assignment of the CA and REC effector domains remains tentative (Supplementary Fig. 5b). However, visual inspection clearly reveals that the dimer interface at the DHp domain is modified in Pfr (Fig. 3a and Supplementary Fig. 5).

Now, we are in a position to compare the Pr and Pfr structures (Fig. 4). A strikingly similar positioning of the PHY domains in Pr and Pfr is observed, despite the change in the fold of the PHY tongue. The distance between the center of mass of PHY domains in Pfr is only 0.4 Å larger compared to the same measure in Pr (Fig. 4a). Yet, a clear change is observed in the densities of the output domains (Figs. 2a and 3a). The change is apparent in the neck region, which shows a parallel arrangement of the helical domains in Pr but spreads apart in Pfr by ~4 Å at the top (Fig. 4b). The two helices of the coiled-coil neck are held together by hydrophobic packing interactions and hydrogen bonds. Notable hydrophobic interactions are shared by Leu502, Leu506, and Ile509 in the lower part of the neck. Asn513, Ser519, and Asn520 form hydrogen bonds across the dimer (Fig. 4c, insets). In the Pfr state, the dimer interface is broken C-terminal of Asn513, which leads to the apparent unzipping of the coiled-coil neck under red light.

## Discussion

From the single particle cryo-EM structures of full-length *Dr*BphP in Pr and Pfr, the following model of photoactivation arises: (1) Red light leads to isomerization of the D-ring in BV. This triggers a change in the chromophore binding pocket and a transition of the PHY tongue from a β-sheet into an α-helix[13]. (2) The PHY domains do not splay apart in a

dramatic way, as has been suggested by the crystal and solution structures of truncated *Dr*BphP[13]. Instead, we find that these changes are much smaller in the context of the full-length protein. (3) The neck opens up in a zipper-like fashion in Pfr, (4) introducing conformational flexibility in the DHp and CA output domains.

In consequence, the dimer interface across the DHp domains breaks, which is consistent with a chromatography study of *Dr*BphP[37]. The change of the shape of the protein is moderate, which is consistent with small-angle X-ray scattering data on the same phytochrome and pulsed electron paramagnetic resonance spectroscopy on a homologous bacterial phytochrome[38,39]. Our structures contradict a previously suggested model of photoconversion based on low-resolution cryo-EM structures[36]. Although the zipper-like opening does not imply a change-of-register in the dimer binding interface of the linker as it has been proposed for phytochrome based on analysis of a phytochrome-regulated diguanylyl cyclases[34], it is still consistent with heptad periodicity of buried residues in the dimer interface[40].

We evaluated the applicability of the mechanism in cyanobacterial, fungal, and plant phytochromes by generating homology models of our Pr structure with the phytochromes from *Synechocystis* PCC6803 (Cph1), *Emericella nidulans* (*En*Phy), and *Arabidopsis thaliana* (*At*PhyA)[41], respectively (Supplementary Fig. 3). We find that all of these phytochromes show potential for a homologous neck region and thus the mechanism, or a variation thereof, may also apply for other phytochromes. However, plant phytochromes signal through light-dependent protein–protein interactions and have recently been shown to have a different dimer arrangement compared to bacterial phytochromes[14]. Other mechanisms of signal transduction may therefore apply in plant phytochromes.

Several crystal structures of truncated phytochromes have both head-to-head and head-to-tail dimer arrangements. In Pr, a bent conformation of the photosensory core module is most often found, whereas varying arrangements have been observed in Pfr[10,11,13,42–44]. These structures may be affected by packing artefacts in the crystal and by the truncation of the phytochrome. The cryo-EM structures presented here are free of these biases. They firmly confirm a head-to-head arrangement of the dimer for bacterial phytochromes and indicate that the bent conformation of the photosensory core module previously detected in Pr crystals is the one that is present in the

**Table 1 | Cryo-EM data collection, refinement and validation statistics**

| | *Dr*BphP(PSM) in Pr (EMD-15684) (PDB ID 8AVV) | *Dr*BphP in Pr (EMD-15685) (PDB ID 8AVW) | *Dr*BphP in Pfr (EMD-15686) (PDB ID 8AVX) |
|---|---|---|---|
| **Data collection and processing** | | | |
| Magnification | 105k | 105k | 105k |
| Voltage (kV) | 300 | 300 | 300 |
| Electron exposure (e⁻/Å²) | 48.3 | 48.3 | 48.3 |
| Defocus range (-µm) | 0.6–2.6 | 0.6–2.6 | 0.6–2.6 |
| Pixel size (Å) | 0.8617 | 0.8617 | 0.8617 |
| Symmetry imposed | C1 | C1 | C1 |
| Initial particle images | 1,051,429 | 1,051,429 | 1,697,254 |
| Final particle images | 117,293 | 117,293 | 98,050 |
| Map resolution overall (Å) | 3.39 | 3.62 | 3.53 |
| FSC threshold | 0.143 | 0.143 | 0.143 |
| Map resolution range (Å) | 2.5–7.0 | 2.5–7.0 | 2.5–7.0 |
| **Refinement** | | | |
| Initial model used (PDB code) | 4Q0J | 4Q0J | 4Q0J + 5C5K |
| Model resolution overall (Å) | 3.39 | 3.62 | 3.53 |
| FSC threshold | 0.143 | 0.143 | 0.143 |
| Model resolution range (Å) | 2.5–7.0 | 2.5–7.0 | 2.5–7.0 |
| Map sharpening *B* factor (Å²) | −86.7 | −84.3 | −77.3 |
| Model composition | | | |
| Non-hydrogen atoms | 7794 | 8992 | 7427 |
| Protein residues | 1010 | 1156 | 961 |
| Ligands | 2 | 2 | 2 |
| B factors (Å²) | | | |
| Protein | 81.5 | 174.1 | 110.5 |
| Ligand | 84.0 | 135.2 | 90.8 |
| R.m.s. deviations | | | |
| Bond lengths (Å) | 0.003 | 0.003 | 0.003 |
| Bond angles (°) | 0.951 | 0.904 | 0.895 |
| Validity | | | |
| MolProbity score | 1.8 | 1.7 | 1.7 |
| Clashscore | 1.8 | 1.8 | 1.6 |
| Poor rotamers (%) | 3.5 | 3.1 | 3.3 |
| Ramachandran plot | | | |
| Favored (%) | 95.1 | 95.1 | 95.2 |
| Allowed (%) | 4.7 | 4.7 | 4.5 |
| Disallowed (%) | 0.2 | 0.2 | 0.2 |

solution. Surprisingly, we found that the same bent arrangement of the module holds for both the Pr and Pfr states.

Like most crystal structures, our cryo-EM models are homodimeric with respect to Pr and Pfr states, and Pr/Pfr heterodimers were not detected. This may be caused by the particle selection process in the EM data processing being unable to assign a class on its own for the Pr/Pfr heterodimers. We believe that the algorithm and data should be accurate enough to do so and have allowed this by requesting more than three classes out of ab-initio refinement and by not imposing any symmetry in the refinement. Thus, the lack of Pr/Pfr heterodimers may

indicate their absence on the grids. This could be explained by fast Pr/Pfr heterodimer reversion to the Pr state during blotting, or that the quantum yield for the photoreaction from Pr to Pfr is allosterically increased in the heterodimer. However, these explanations remain somewhat speculative and we believe that the issue requires further investigation.

The structures are the first full-length and dimeric structures of a TCS histidine kinase in active and resting states. Firstly, the structures show evidence of a zipper-like mechanism to guide signal transduction into the effector domains. To the best of our knowledge, this is the first observation of such a mechanism and adds to earlier proposals for various histidine kinase architectures on scissoring piston-like movements, or rotation of the helical bundles[21,23]. Several mechanisms of signal transduction into the effector domains may coexist[25], reflecting that histidine kinases integrate many different signals in a variety of architectures and have evolved through domain swapping[45]. Similarly, the photosensory modules of phytochromes may adapt their structural changes to the output domains, which may explain why they can form a variety of domain fusions with distinct effectors[16,17,40,46]. Secondly, our data provide direct and strong evidence that the effector histidine kinase module loses symmetry and order when its biochemical activity changes. Crystal structures of enzymatically active HKs do not show disorder, which is however likely due to that diffracting crystals require ordered packing. The present cryo-EM structures are free of such restrictions representing the structure of the proteins in solution much more closely.

Finally, our structures provide insight into how allosteric communication across the entire bacteriophytochrome occurs. We observe structural changes around the chromophore and in the output domains. Apart from the secondary structural changes of the tongue region, the PHY domains, which connect the chromophore region to the effector domains, undergo only a rather small change of position. The center of mass of the PHY domains moves apart by only 0.4 Å in Pfr compared to Pr. We conclude that the small structural changes in the PHY domains are sufficient to tip the scales and break the dimer interface across the neck, which then controls the enzymatic activity of the effector domain. The dimer interface at the DHp domains thereby amplifies the structural changes in the PHY domain and plays a key part in signaling along the phytochromes. This interpretation is in line with allostery where coupled equilibria between domains transduce information across the entire phytochrome[25,28]. Realizing that coupling across the entire phytochrome exists—and persists after photoexcitation—provides a structural rationale for that the dark reversion time back to the resting state changes when the protein is truncated at the neck[30].

The structures of a prototypical full-length bacteriophytochrome in active and resting states reveal a zipper-like mechanism of how the signal is transduced into the effector domains in phytochromes, explaining how bacteria, plants, and fungi sense red/far-red light. This mechanism serves as a template for understanding two-component signaling and can inspire the development of new near-infrared optogenetic applications[16,17,47].

## Methods

### Protein preparation and purification

The phytochrome from *Deinococcus radiodurans* strain R1 (*Dr*BphP, gene DR_A0050) in pET21b(+) plasmid (Novagen) was a kind gift from Prof. Richard Vierstra[48]. The response regulator from *Deinococcus radiodurans* strain R1 (*Dr*RR, gene DR_A0049) is described elsewhere[35]. The *Dr*BphP-*Dr*RR fusion protein was generated by introducing the *Dr*RR gene (residues 1–149) after *Dr*BphP (residues 1–755) and before the XhoI restriction site of the pET21b(+) with Gibson assembly (NEBuilder HiFi DNA assembly cloning kit, New England Biolabs). A linker of 12 residues (ASSAGGSAGSAG), inspired by the Agp2 sequence, was also introduced between the *Dr*BphP and *Dr*RR. The resulting 938-aa

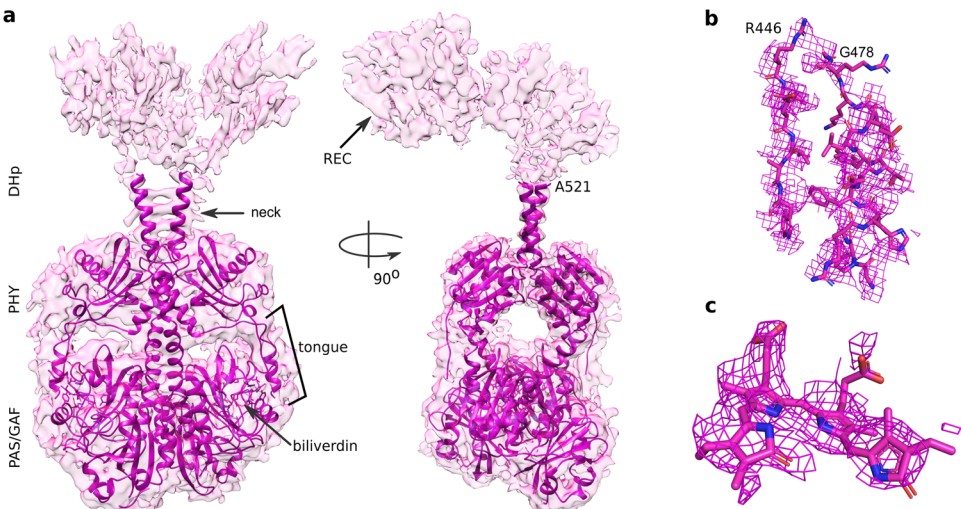

**Fig. 3 | The structure of *Dr*BphP-*Dr*RR in Pfr. a** The reconstructed electron density of Pfr full-length is shown with the structural model, which was refined up to residue Ala521 based on the cryo-EM map of the photosensory core module in Pfr (Pfr PSM). Patches of electron density were assigned to the CA and REC domains (see Supplementary Fig. 5b). **b** The tongue is in α-helical form characteristic of Pfr-state phytochrome structures. **c** The density of the BV chromophore supports a 15*E* conformation (compare to Supplementary Fig. 5d). The densities of the PHY tongue and BV are extracted from the cryo-EM map with the local refinement of the PSM and neck in Pfr (Pfr PSM, see Supplementary Fig. 4).

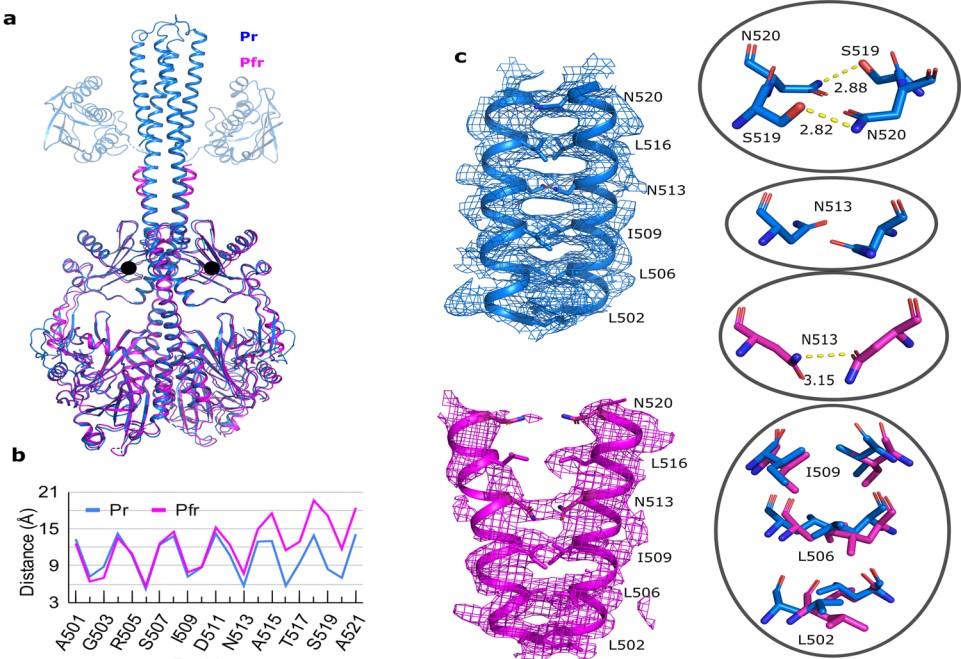

**Fig. 4 | Comparison of *Dr*BphP-*Dr*RR in Pr (blue) and Pfr (magenta) reveals a zipper-like opening at the neck. a** The entire structure is compared. The CA domains in the Pr-state structure are shown as transparent. They were not refined against the data and are taken from a homology model against the structure of a histidine kinase from *Thermotoga maritima* (pdb entry 2C2A)[52]. The black dots indicate the center of mass of the PHY domains. **b** The graph shows the distance between Cα atoms across the dimer interface in the neck region. Source data are provided as a Source Data file. **c** The neck region with the densities is shown for Pr (blue) and Pfr (magenta). A zipper-like opening of the dimerization interface is observed. The insets show detailed interactions between sister residues Ser519 and Asn520 (upper), Asn513 (middle), and Leu502, Leu506 and Ile509 (lower). The structures are colored according to their state, the hydrogen bonds are shown in yellow dashed lines, and the distances are in Å.

construct had a PAS-GAF-PHY-DHp-CA-REC domain composition flanked by an N-terminal T7-tag and a C-terminal His₆-tag.

*Dr*BphP-*Dr*RR and *Dr*BphP were expressed and purified as described below and adapted from ref. 35. The expression was conducted in *Escherichia coli* strain BL21 (DE3) overnight at 24 °C. After cell lysis with EmulsiFlex®, the sample was incubated with a molar excess of biliverdin hydrochloride (Frontier Scientific) overnight on ice. The His₆-tagged protein was purified with NiNTA affinity purification (HisTrap™, GE Healthcare), followed by size-exclusion chromatography (HiLoad™ 26/600 Superdex™ 200 pg, GE Healthcare) in buffer (30 mM Tris, pH 8.0). Finally, the purified protein was concentrated to 12 mg/ml and flash-frozen. Prior to application to the grids, the protein

was thawed and purified by size exclusion chromatography (Supplementary Fig. 1a) to ensure sample homogeneity.

## UV–vis spectroscopy and Phos-tag assay

The absorption spectra of phytochromes were measured with Agilent Cary 8454 UV–visible spectrophotometer (Agilent). Phytochrome samples were diluted with (30 mM Tris/HCl pH 8.0) to 1.0 μM concentration and an approximate $A_{700}$ value of 0.1 cm$^{-1}$. The samples were illuminated with saturating (662 ± 1) nm red light and (782 ± 1) nm far-red light to populate Pfr and Pr states, respectively. In the dark reversion experiments, phytochromes were first driven to the Pfr-state population with 3-min saturating red light, after which the reversion data were recorded at room temperature. Data points were taken at 2-min intervals for the first 10 min, 5-min intervals until 30 min, 10-min intervals until 60 min, and 20-min intervals until 120 min. The exponential fits from dark reversion data were calculated with Matlab R2020b (9.9.0.1467703) (MathWorks Inc.) using Eq. (1). Two components were used for fitting in all samples.

$$\frac{A_{750}}{A_{700}}(t) = A_1 e^{-\frac{t}{\tau_1}} + A_2 e^{-\frac{t}{\tau_2}} \tag{1}$$

where $t$ is time, $A_{700}$ and $A_{750}$ are absorption values, $A_1$ and $A_2$ are the decay amplitudes of the absorbance ratio, and $\tau_n$ the time constant of the decay component.

The Phos-tag assay was carried out with a similar method as in Ref. 3. Namely, phosphorylated *Dr*RR (p-*Dr*RR) was generated by incubating *Dr*RR in the assay buffer (25 mM Tris/HCl pH 7.8, 5 mM MgCl$_2$, 4 mM 2-mercaptoethanol, 5% ethylene glycol) supplemented with 90 mM acetyl phosphate at 37 °C for 30 min. For the phosphatase reactions in the assay buffer, p-*Dr*RR (0.3 mg/ml) was mixed with phytochrome samples (0.3 mg/ml each), and the reactions were initiated with 1.67 mM ATP. The reactions were incubated either in dark or under saturating 650-nm light (LED Spot Light, 3 W Deep Red 657 nm, Lens-C) for 20 min after which the reaction was stopped and analyzed according to Zn$^{2+}$-Phos-tag® SDS–PAGE assay (Wako Chemicals)[35].

## Single-particle cryo-EM grid preparation and data acquisitions

Cryo-EM grids were prepared under dim green safe light using a Vitrobot (FEI) with the sample chamber at 4 °C and 100% humidity. Before grids preparation, CaCl$_2$, MgCl$_2$, and AMP-PNP were added to the sample to 10, 10, and 4 mM concentrations, respectively. To prepare grids with proteins in Pr state, three microliter protein samples (1–1.5 mg/ml) were applied to glow-discharged Quantifoil R 2/2 Cu300 grids and pre-illuminated with 780 nm far-red light for Pr state and 660 nm red light for Pfr state on the grid before plunge-freezing in liquid ethane. The particles were imaged using a Titan Krios operated at 300 kV and a magnification of 105k. The images were recorded on a Gatan K3 BioQuantum detector with a pixel size of 0.8617 Å and an exposure rate of 48.3 electrons per Å$^2$ for a total of 40 frames. The targeted defocus range was varied from −0.6 to −2.6 μm using the EPU software (Thermo Fisher).

## Analysis of cryo-EM data

A total of 6392 movies were collected from a grid prepared in far-red light illuminated (Pr) state and 7251 movies from a grid prepared in red light illuminated (Pfr) state, see Supplementary Fig. 2 and Supplementary Fig. 4 for the scheme of image processing. Movies (Supplementary Fig. 8a, b) were motion-corrected and contrast transfer function (CTF)-estimated using cryoSPARC v3[49]. After blob-picker, extracted particles were subjected to 2D classification to remove the junk particles. Good particles were used in Topaz deep pick[50]. Extracted particles were further cleaned up with more rounds of 2D classification (Supplementary Fig. 8c, d). From a far-red

light-illuminated grid, a set of 316,139 particles was used to reconstruct three initial 3D models using Ab-initio in cryoSPARC. These three models were used as initial volumes for heterogeneous refinement and the class 1 model with 117,297 particles showed the best features and was chosen to further homogeneous and local refinement without any symmetry applied. The final model of the Pr state has an overall resolution of 3.6 Å (Supplementary Fig. 6). From the grid prepared in the red light illuminated state, good particles with side view only from the blob-picker went through an extra round of Topaz deep pick. After duplicated particles got removed, a set of 387,312 particles was used to reconstruct three initial 3D models. After heterogeneous refinement, the class 2 model was identified to be Pfr state and the class 1 model Pr state. After two more rounds of heterogeneous refinement to clean up particles of Pr state from Pfr state, a total of 98,050 particles were used for further homogeneous, local refinement, and sharpening.

## Model building and refinement

The crystal structure of *Dr*PAS-GAF-PHY fragment (pdb code 4Q0J) was first fit into the Pr full-length map (Pr full-length, see Supplementary Fig. 2), then built and refined further in real space with Coot 0.9.6. We were able to build the *Dr*BphP up to residue Leu591, covering the PAS-GAF-PHY-DHp domains. The Pfr structure was initiated by fitting the *Dr*PAS-GAF-PHY crystal structure (pdb code 4Q0J) into the map of the photosensory core module (Pfr PSM, see Supplementary Fig. 4), followed by modeling the Pfr-specific traits at the chromophore surroundings and the PHY tongue based on a Pfr-state structure of *Dr*PAS-GAF-PHY (pdb code 5C5K). The model was built up to Ala521, covering the neck part of the DHp. Due to the poor density in the histidine kinase module, model building was not conducted with the Pfr full-length map (Pfr full-length, see Supplementary Fig. 4). Once the Pr and Pfr models were built and refined, the structures were refined with REFMAC5 (version 5.8.0267) of the CCP-EM software suite (version 1.5.0) with tight (0.0002–0.001) manual weights[51]. The model and refinement parameters are collected in Table 1.

## Reporting summary

Further information on research design is available in the Nature Portfolio Reporting Summary linked to this article.

# Data availability

The data that support this study are available from the corresponding authors upon reasonable request. The cryo-EM maps have been deposited in the Electron Microscopy Data Bank (EMDB) under accession codes EMD-15684 (DrBphP(PSM) in PR), EMD-15685 (DrBphP in Pr), and EMD-15686 (DrBphP in Pfr). The coordinated have been deposited in the Protein Data Bank (PDB) under accession codes 8AVV (DrBphP(PSM) in PR), 8AVW (DrBphP in Pr), and 8AVX (DrBphP in Pfr). Source data are provided with this paper.

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

## Acknowledgements

Cryo-EM samples were prepared at the Centre for Cellular Imaging at the University of Gothenburg. Cryo-EM sample screening and data collection were performed at the cryo-EM Swedish National Facility in Stockholm, Sweden funded by the Knut and Alice Wallenberg, Family Erling Persson and Kempe Foundations, SciLifeLab, Stockholm University, and Umeå University. We thank Julian Conrad, Karin Walldén, Michael Hall, and Marta Carroni for the technical assistance in data collection and the advice on the reconstruction of the electron density maps. We thank Dr. Dmitry Morozov (University of Jyväskylä) for help in modeling the structure and fruitful discussions, as well as Vilma Tarkiainen and Roosa Vanhatalo (University of Jyväskylä) for help with the sample preparation. S.W. acknowledges support from the European Research Council, ERC grant to Westenhoff, MolStrucDyn, 725642. This work was supported by Academy of Finland grants 332742 (J.A.I.) and 330678 (H.T.).

## Author contributions

S.W., J.A.I., H.T., and E.C. perceived the project; S.W., W.Y.W., E.C., H.T., and J.A.I. devised the methodology; W.Y.W., S.T.-M., and I.T. performed the investigation; W.Y.W., S.B., and H.T. prepared the figures; S.W., H.T., and J.A.I. secured the funding and supervised the project; S.W., H.T., S.B., and W.Y.W. wrote the paper with input from all authors.

## Funding

## Competing interests

The authors declare no competing interests.
