## [Peer Review File · Nature Communications]

Structural mechanism of signal transduction in a phytochrome histidine kinaseReviewers' Comments:

Reviewer #1:

Remarks to the Author:

The present work by Wahlgren et al expands on the group's investigation on the structure and activation mechanism of a bacteriophytochrome from *Deinococcus*. The main data presented here are a pair of *Deinococcus* phytochrome structures determined by cryo-electron microscopy as Pr (the dark resting state) and Pfr (the illuminated state). The nature of the data presented here is in principle similar to that by Li et al 2010 (doi/10.1073/pnas.1001908107), but the authors take advantage of the recent technical advance in cryo-electron microscopy to determine structures of *Deinococcus* phytochrome in both Pr and Pfr states, successfully assigning atomic coordinates to much of the protein.

The protein was utilised as a chimeric fusion comprising *Deinococcus* phytochrome and DrRR, the cognate response regulator. The determined structure models comprise the nPAS-GAF-PHY-neck-DHp regions and nPAS-GAF-PHY-neck regions for Pr and Pfr structures, respectively. While neither Pr and Pfr structures have been modelled with the complete protein moiety of the chimeric construct (particularly the CA and RR domains), the revealed quaternary structure at the PSM is expected to be more biologically relevant than the group's previously published pair of *Deinococcus* phytochrome crystal structures (Takala et al 2014, 4O0P and 4O0I) for the following reasons:-

1. The construct encompasses the full-length *Deinococcus* phytochrome (instead of only PSM, nPAS-GAF-PHY).
2. Cryo-EM captures protein structures in solution free from crystal contact, therefore the protein conformations likely represent their native states.

The quaternary structure of phytochrome has been the topic of much discussion in the field due to its implication in signal transduction from the N-terminal sensory region to the C-terminal signalling region. Therefore, this new pair of phytochrome cryo-EM structures merits publication, not only for the new information presented in the manuscript, but also the potential for the field to utilize the structural information for downstream applications such as molecular dynamic simulation or construction of more accurate homology models of full length prokaryotic phytochromes.

P11 line 21

... with phytochrome-like ...

As the first example of a prokaryotic phytochrome with the canonical domain organisation (<https://doi.org/10.1038/386663a0>), the description of Cph1 (a bona fide phytochrome) as being "phytochrome-like" is inappropriate.

P10 line 11 onwards & p11 line 9: On the discussion on the inter-molecule distance between Pr and Pfr. An EPR study on full-length Agp1 has been published, which reports that the inter-molecular distance between Agp1 517th residues (close equivalent of DrBphP N513) do not change radically between Pr and Pfr. This EPR data supports the manuscript's findings based on cryo-EM structures, which contrast to the splayed apart PHY domains of the Pfr crystal structure.

Kacprzak et al 2017 JBC, DOI 10.1074/jbc.M116.761882

P11 line 22 and Fig. S3

A signal transduction model involving the opening of the neck region is described as being applicable to other phytochromes, including plant phytochrome A. There are two reasons that argues against this.

1. The homology model of AtPhyA presented in Fig. S3 does not resemble the cryo-EM structure of the same protein presented by Wahlgren et al 2021 (doi: 10.3389/fpls.2021.663751), particularly regarding the position of the PAS-A domain and the "neck", where the latter is an α -helical region implicated in the mechanism of signal transduction the present manuscript. While the structure presented in 2021 is at the resolution of 17 Å, it is an empirical experimental data. How unexpected

the real structure can be (i.e. how wrong the homology model can be) was recently exemplified by the cryo-EM structure of AtphyB (Li et al 2022 <https://doi.org/10.1038/s41586-022-04529-z>).

2. Many bacteriophytochromes function as histidine kinase / phosphatase by the virtue of the C-terminal HK region, whereas plant phytochromes signal predominantly by protein-protein interaction (& its capacity as a kinase at the C-terminus is under dispute). Additionally, plant phytochromes feature two PAS domains between the N-terminal sensory region and the C-terminal HK-related region. Given that the present manuscript is implicating the neck opening to the control of the C-terminal enzyme activity, it seems far-fetched to apply the proposed signal transduction model to plant phytochromes without additional evidence.

On the similar note, by referring to "plants" (p13 line 5) it implies to include phyB, which seem inappropriate. Because phyB was recently shown to feature an unexpected quaternary structure (Li et al 2022, anti-parallel at the PSM with PAS2s in between, parallel at the HK-related): Therefore the opening of the neck model will not apply to phyB, at least not within the same framework described in the present manuscript.

Structural information presented in the present manuscript isn't necessarily incompatible with the picture presented by Multäki et al 2021 (<https://doi.org/10.1038/s41467-021-24676-7>), i.e. DrBphP interacts with DrRR even in Pfr (although slightly weaker than when in Pr). (e.g. It could be that in Pfr the DHp & CA swings about while retaining interaction with DrRR, hence weaker density map) What is not explained, however, is how the break of the DHp interface and rise of asymmetry at the C-terminus in Pfr lead to higher phosphatase activity (lower kinase activity), while more stable, symmetric DHp interface leads to lower phosphatase activity (higher kinase). One imagines both phosphatase and kinase activities require interaction between HK and RR. Would it be possible to comment on this?

Although the authors are now in a good position to describe the tertiary structure/inter-subunit interface at the PSM in a more biologically relevant setting (in contrast to crystal structures of fragments), I feel description in this area is scarce. It would be beneficial to describe this aspect, if possible (without going over the word/page limit specified by Nat. Comm.). Many crystal structures of all BphPs (with canonical domain organisation) have been determined as fragments. Therefore the correlation observed in the past (Pr: bent/kinked overall PSM conformation, Pfr: more straight conformation) wasn't definitive (fragment construct/crystal packing). This new pair of DrBphP cryo-EM structures overcome that limitation. This is perhaps not unimportant since the change in dimer interface is what causes the breaking of the DHp interface.

Supplementary document, Fig. S2 and S4: On the particle selection and refinement process.

There are 3 classes of maps generated each for Pr and Pfr following the ab initio reconstruction. It seems arbitrary to choose one class for both Pr and Pfr, which were then refined and used as a basis of discussion. But the %representation was not the criteria for choosing the class for subsequent refinement and discussion. E.g. For Pr, the class 2 represents higher proportion (41.8%) than that of class I (subsequently refined, 37.1%).

For Pfr: Ignore the class 1 (20.0%) since it represents Pr: class 3 (48.3%) represents higher proportion than class 2 (subsequently refined, 31.7%).

When protein conformation (and difference between Pr & Pfr) is under discussion, it appears strange to take minority conformation species for both Pr and Pfr as a basis of discussion, especially when classes show very different conformations (apparently much more so than for example a case of AtphyB by Li et al 2022). Shouldn't the variety of conformation be part of the discussion too?

A related question to the above topic

Are there any possibility that some of the particles in the Pfr grid represented Pr/Pfr heterodimer?

P5, Fig. 1 Legend: The effect of chimeric fusion on the phosphatase activity,
... because of competition between the fused and free DrRR.

There is no evidence that the mode of interaction between DrBphP and DrRR in the context of chimeric fusion dimer protein is the same as that of the inter-molecular interaction between two separate proteins. From the practical point of view, we may not need to understand the reason why the chimeric fusion strategy led to stabilization of the protein to render it suitable for cryo-EM. From the point of view of those wishing to learn from this manuscript, however, it will be useful to know whether it was indeed the biological interaction which was the key to stabilize the sample for cryo-EM. While not essential, it will be preferable if this question is addressed (although it will be difficult to discern between intra- and inter-molecular interaction).

P10 line 14 and p12 line 18

How the value of 0.4 Å was derived isn't clear. (e.g. If two PHY domains (only) are superposed with one another and their centres of mass are compared, the distance between the centres will naturally be very small) Was superposition carried out based on the entire PSM or only nPAS-GAF?

P3 Line 20

... in light-activated and resting states 7-9

Essen et al 2008, Yang et al 2008, and Wagner et al 2005 (2VEA, 3C2W, and 1ZTU) are all in resting states. (PaBphP is a bathy-phytochrome, Pr is its activated state, for both photoconversion or HK activity)

P4 line 5

... daft of structures ...

"Daft" sounds inappropriate, could you please check? Dearth? But even the latter isn't so common.

P16 line 23 & p17 line 3, and Table S1: Initial model used

4Q0J (DrBphP in Pr) is consistently described as the initial model. Was this also true for the cryo-EM structural determination for DrBphP in Pfr? One imagines either 4O01 or 5C5K (DrBphP as illuminated/Pfr) are more suitable as initial models.

Figure S3 Legend: On the difference between cryo-EM and crystal structures (Pfr)

... but that the Pfr crystal structure does not (overlap)

Any two proteins can be superposed with one another, but the question is how good the superposition is. Would it be possible to describe this by quantity? E.g. superposition using the entire PAS-GAF-PHY polypeptide (RMSD). In this case the value calculated for Pfr is expected to be larger than that of Pr.

P12 line 8: it may be that several mechanisms of signal transduction into the effector domains coexist. While this is a simple explanation, the picture becomes more complicated when one considers past studies involving a chimera based on Cph1 phytochrome (Levskaia et al 2005, <https://doi.org/10.1038/nature04405>). This chimeric enzyme (PSM of Cph1 and HK of EnvZ) exhibit light regulated enzyme activity, therefore the mechanism of signal transduction appears to be the same (or at least compatible) between Cph1 and EnvZ. EnvZ is a HK which features a HAMP domain, and HAMP domain was proposed to transmit signal via rotation (Hulko et al 2006, DOI 10.1016/j.cell.2006.06.058). This presents a strange picture where Cph1 (argued to operate via the neck opening in the present manuscript) sensory module is working with EnvZ (argued to work via the rotational model). Does this mean Cph1 can both break at the neck and rotate upon Pr/Pfr? Other similar studies utilising HK chimeras exist. This probably connects to modularity of HK domains and evolution of HK via domain swapping in prokaryotes. While one appreciates the difficulty in covering this topic extensively, the discussion will become more informative by expansion, if possible.

Reviewer #2:

Remarks to the Author:

This paper describes the cryoEM structures of the full-length *Deinococcus radiodurans* bacteriophytochrome (DrBphP) in both the Pr and Pfr forms. This molecule has served as a key subject for studying many aspects of phytochrome light activation and signaling; understanding how it functions as a full-length protein, i.e. how the output signal is converted through its many domains to the enzymatic output is an important goal that will impact many fields, including biophysics, structural biology, signal transduction, photobiology and even plant biology, given the critical role of plant phytochromes in regulating growth and development. The key finding is that in the Pfr form, cofactor isomerization and structural rearrangement of the tongue element (which had been previously characterized) lead to an “unzipping” of the neck helices, that then causes increased conformational disorder and asymmetry (or dynamics) in the histidine kinase (CA) and receiver (REC) modules. Given that this is a full-length, multidomain signaling system, these results add substantially to what we understand about sensor histidine kinase regulation and complements the insights from previous studies. Unfortunately, the resolution and local quality of the Pfr structure is not sufficient to provide much more molecular detail on the structural changes that are propagated from the cofactor to the neck, as the Phy domain itself, which contacts the neck, changes minimally within the resolution of the structures. Overall, I do think this is an important work, but I do have some serious concerns, some technical, and some with interpretation, as described below:

1. Reconstructions: The quality of the density in the Pfr form, particularly in the neck, and also in the CA/receiver regions appears skewed and streaked, a possible consequence of orientational bias or even mixed classes. Orientational bias may indeed be indicated suggested by the FSC curves and the unusual bump around 5 Å resolution. The orientational sampling data for the structures is not provided, and should be. Whereas there clearly seems to be increased disorder in the Pfr state, there is some question as to whether the unzipping of the helices can be solidly interpreted. It’s also unclear as to whether the structure is truly more asymmetric than the Pr state, or just more disordered. Only a preliminary validation report is provided for the structures. Are there problems that prevented submission of the finalized report? In that regard, the stereochemical statistics of the two models are somewhat worrisome, especially the percentile of the clashscore and sidechain outliers. The main argument for conformational signaling is heavily based on the alternative conformations of the zipper-like neck, particularly the distance changes of N513, L516, and N520 which locate on the dimer interface (figure 4); however, the validation reports suggested in both models, that those residues have unfavorable conformations. Also, the zipper region does not fit very well to the density, especially in the Pfr state (e.g. fig. 4b bottom panel, the left alpha-helix looks displaced), which may be caused by the moderate quality of the modeling, again implied by the clashscore. It is convincing that the two models have significantly different conformations that propagate throughout their dimensions, but the quality of the models could be improved so as to avoid unnecessary bias in the interpretation.

2. The biochemical experiments of Fig. 1c are somewhat difficult to interpret, more description in the legend would help (for example, it is not stated that the phosphorylated p-DrRR is added exogenously in the legend). Why is there substantial phosphatase activity of the DrBphP in the D state, nearly as much as in R?

3. The paper sets the context by stating the importance of sensor kinase signaling, and in several places refers to other conformational mechanisms of sensor kinases in comparison. However, DrBphP has no kinase activity, it is a light-activated phosphatase and hence this point should be made in considering any comparison to other histidine kinases. There is the general question of what role the CA domains play here at all and whether a change in their order is relevant to phosphatase activity. It is true that for EnZ the CA domains enhance phosphatase activity of the Dhp domains, is this known for Dr Bphp as well? Overall, the paper should address these mechanistic issues more clearly and in particular how the structural changes observed relate to the key issues of this specific protein’s activity before generalizing to the entire family of sensor kinases.

Line 194 - The work of Reference 26 is an interesting comparison, but more discussion is warranted so as not to be misleading. Although the inhibited state of LuxQ is proposed to be asymmetric, the observed asymmetry occurs in the periplasmic sensing domain. The sensing domain of BpHp, appears to roughly symmetric, as least not nearly as asymmetric as LuxQ, so its unlikely an asymmetric signal on the order of LuxQ could be involved here. Nevertheless, the Pfr structure may provide some evidence for an asymmetric cytoplasmic domain conformation in the phosphatase form, which would indeed be relevant to LuxQ.

Lines 179-180 – more discussion is needed here – given the sentence before, one might conclude that this study contradicts the findings of Reference 34. However, the molecule studied in 34 is a diguanylate cyclase; the output domain is very different than in the phytochrome in that the enzymatic dGC is an obligate dimer and hence this oligomerization will likely influence any conformational changes in the connecting helices. It is a very interesting comparison, but these mechanisms need not be in opposition – they almost have to be different.

Lines 186-190 – with respect to scissoring, it seems to me that density in Pfr compared to Pr could be interpreted as a scissoring motion, or at least have a component of a scissoring motion, I also don't think that helical rotations can be fully ruled out at this resolution, especially given the concerns with the modeling and stereochemistry of these regions, as stated above.

Lines 193-194 – it's clear that order is lost, but less clear that symmetry is also lost, or rather that asymmetry is really a key aspect of the activation mechanism. The density in the validation reports looks reasonably symmetric, perhaps less so than with Pfr, but with a broader conformational ensemble, classification is going to be more difficult (I note that a lot of particles are excluded in the final reconstruction) and the result of a slightly asymmetric looking particle more likely.

Lines 58-62 and later 196-202 – I do think it is a significant finding that there is little change in the phy domains to propagate the signal, but the discussion with respect to different types of allostery could use some more development. Lines 58-62 – both “a cascade of structural changes leading to signaling” and “allosteric coupling between modules” fall under the modern explanation of allostery, which really comes down to coupled conformational changes at separated (binding) sites. There can be many mechanisms, involving a few specific moieties, or a more global “refolding” of domains and these altered interactions can certainly couple across interfaces, but all of these mechanisms can be considered allostery – perhaps more specifics need to be provided to allow the reader to understand better what is being distinguished.

Line 184 and Fig. S3 – *A. thaliana* PhyA is more closely related to *At* PhyB than *DrBpHP*, and the recent cryoEM structure of PhyB shows the neck region to be in a completely different conformation than it is here – it would seem that a homology model for PhyA based on PhyB is more appropriate.

Last sentence – “inspire the development of new antibiotics and new optogenetic tools” – it would be helpful to say a little more about these points. There's really no discussion of pathogenesis in the paper or why phytochromes would be a good target for drugs and how these structures would help with that, likewise for the optogenetic tools.

Data processing – does the class for the Pr state extracted from the red-light-treated ensemble, match that of the pure Pr state?

Figure S5 is quite confusing, and the assignment of the CA and REC not very convincing – was there any computational docking attempted to better validate the placements? Or perhaps attempts at a focused refinement of the map on the area where the domains reside?

Minor:

Abstract "which sensing is archived" is an unfamiliar formulation, consider rewording

Line 50 : perhaps ..."used as" near-infrared fluorescent markers...

Fig.2 – it is unclear what the arrow for the CA domain is pointing at – can the CA and REC domains be distinguished at all? Perhaps brackets would help? I'd also recommend using a different color for the model and the density so that they can be better distinguished – same comment for Fig. 3.

Lines 116-117 – would do you mean by this statement, that the helices of the DHp could be connected in a different handedness? More explanation is needed.

Line 192 – do you mean that multiple mechanisms coexist in for the same protein, or different proteins have different mechanisms?

Fig. 3 – label residue 521 in the models. Part c, given the map at the BV is fairly low resolution a superposition with the 15Z conformation would help convince the reader that the conformation is different.

Line 171 – "dramatical" should be "dramatic"

Line180 – It doesn't seem like the residues on the interior of the helices are still buried in the model? – 516-520 move far enough away to accommodate water, and after that the helices are undefined.

Note that there are a few plural/singular issues and other typos throughout.

REVIEWER COMMENTS

Reviewer #1 (Remarks to the Author):

The present work by Wahlgren et al expands on the group's investigation on the structure and activation mechanism of a bacteriophytochrome from *Deinococcus*. The main data presented here are a pair of *Deinococcus* phytochrome structures determined by cryo-electron microscopy as Pr (the dark resting state) and Pfr (the illuminated state). The nature of the data presented here is in principle similar to that by Li et al 2010 (doi/10.1073/pnas.1001908107), but the authors take advantage of the recent technical advance in cryo-electron microscopy to determine structures of *Deinococcus* phytochrome in both Pr and Pfr states, successfully assigning atomic coordinates to much of the protein.

The protein was utilised as a chimeric fusion comprising *Deinococcus* phytochrome and DrRR, the cognate response regulator. The determined structure models comprise the nPAS-GAF-PHY-neck-DHp regions and nPAS-GAF-PHY-neck regions for Pr and Pfr structures, respectively. While neither Pr and Pfr structures have been modelled with the complete protein moiety of the chimeric construct (particularly the CA and RR domains), the revealed quaternary structure at the PSM is expected to be more biologically relevant than the group's previously published pair of *Deinococcus* phytochrome crystal structures (Takala et al 2014, 4O0P and 4O0I) for the following reasons:-

1. The construct encompasses the full-length *Deinococcus* phytochrome (instead of only PSM, nPAS-GAF-PHY).
2. Cryo-EM captures protein structures in solution free from crystal contact, therefore the protein conformations likely represent their native states.

The quaternary structure of phytochrome has been the topic of much discussion in the field due to its implication in signal transduction from the N-terminal sensory region to the C-terminal signalling region. Therefore, this new pair of phytochrome cryo-EM structures merits publication, not only for the new information presented in the manuscript, but also the potential for the field to utilize the structural information for downstream applications such as molecular dynamic simulation or construction of more accurate homology models of full length prokaryotic phytochromes.

We thank the reviewer for the review and this positive summary. We have addressed the comments below and believe that this has increased the quality of the paper notably.

P11 line 21

... with phytochrome-like ...

As the first example of a prokaryotic phytochrome with the canonical domain organisation (<https://doi.org/10.1038/386663a0>), the description of Cph1 (a bona fide phytochrome) as being “phytochrome-like” is inappropriate.

We agree, and have adjusted the sentence.

P10 line 11 onwards & p11 line 9: On the discussion on the inter-molecule distance between Pr and Pfr

An EPR study on full-length Agp1 has been published, which reports that the inter-molecular distance between Agp1 517th residues (close equivalent of DrBphP N513) do not change radically between Pr and Pfr. This EPR data supports the manuscript’s findings based on cryo-EM structures, which contrast to the splayed apart PHY domains of the Pfr crystal structure. Kacprzak et al 2017 JBC, DOI 10.1074/jbc.M116.761882

We thank the reviewer for reminding us about this paper. At the time it was surprising to find almost no changes in the EPR distances, but it agrees well with the findings in our current investigation. The paper is now acknowledged in agreement of the cryo EM models. (page 12)

P11 line 22 and Fig. S3

A signal transduction model involving the opening of the neck region is described as being applicable to other phytochromes, including plant phytochrome A. There are two reasons that argues against this.

1. The homology model of AtphyA presented in Fig. S3 does not resemble the cryo-EM structure of the same protein presented by Wahlgren et al 2021 (doi: 10.3389/fpls.2021.663751), particularly regarding the position of the PAS-A domain and the “neck”, where the latter is an α -helical region implicated in the mechanism of signal transduction the present manuscript. While the structure presented in 2021 is at the resolution of 17 Å, it is an empirical experimental data. How unexpected the real structure can be (i.e. how wrong the homology model can be) was recently exemplified by the cryo-EM structure of AtphyB (Li et al 2022 <https://doi.org/10.1038/s41586-022-04529-z>).

2. Many bacteriophytochromes function as histidine kinase / phosphatase by the virtue of the C-terminal HK region, whereas plant phytochromes signal predominantly by protein-protein interaction (& its capacity as a kinase at the C-terminus is under dispute). Additionally, plant phytochromes feature two PAS domains between the N-terminal sensory region and the C-terminal HK-related region. Given that the present manuscript is implicating the neck opening to the control of the C-terminal enzyme activity, it seems far-fetched to apply the proposed signal transduction model to plant phytochromes without additional evidence.

On the similar note, by referring to “plants” (p13 line 5) it implies to include phyB, which seem inappropriate. Because phyB was recently shown to feature an unexpected quaternary structure (Li et al 2022, anti-parallel at the PSM with PAS2s in between, parallel at the HK-related): Therefore the opening of the neck model will not apply to phyB, at least not within the same framework described in the present manuscript.

We agree with the reviewer - in light of the new structure on plant PhyB by Li *et al.* 2022 and due to that plant phytochromes mainly use protein-protein interactions to transducing signals, the neck-zipper mechanism is likely not the only one that is active in plant phytochromes. We have now included an AlphaFold model (with AlphaFold multimer) of AtPhyA in the Supplementary Information. In all plotted homology/AlphaFold models represented in Fig. SI3, the neck region is present and provides for buried interface so it is possible that opening of the neck transduce a signal to the histidine kinase-like C-terminal domains.

We have changed the SI figure and have adjusted the referring text in the discussion section to clarify this discussion point.

Structural information presented in the present manuscript isn't necessarily incompatible with the picture presented by Multäki et al 2021 (<https://doi.org/10.1038/s41467-021-24676-7>), i.e. DrBphP interacts with DrRR even in Pfr (although slightly weaker than when in Pr). (e.g. It could be that in Pfr the DHp & CA swings about while retaining interaction with DrRR, hence weaker density map)

What is not explained, however, is how the break of the DHp interface and rise of asymmetry at the C-terminus in Pfr lead to higher phosphatase activity (lower kinase activity), while more stable, symmetric DHp interface leads to lower phosphatase activity (higher kinase). One imagines both phosphatase and kinase activities require interaction between HK and RR. Would it be possible to comment on this?

The findings in the Multimäki et al. 2021 paper rest on a comparison of *DrBphP* (phosphatase activity in Pfr) and *Agp1* (histidine kinase activity in Pr). We have considered several models to explain this in light of the new structures presented in the present paper. We agree with the reviewer that in both cases (phosphatase and kinase activity) the phytochrome structure should support transient contact with its cognate RR.

In light of the results here and the ones in Multamäki et al. 2021, it appears that asymmetry or disorder in the Pfr state favors *DrBphP* phosphatase activity. One can speculate that this higher disorder/asymmetry is related to the increased enzymatic activity. This for example would mean in the case of *Agp1* that its Pr-state structure would be more disordered and hence supporting kinase activity. Alternatively, structural disorder may be characteristic of phosphatase activity of the HKs and more rigid structure of kinase activity. However, these presented ideas remain fully speculative until more full-length HK structures have been identified.

We have now included discussion about the the relation of HK (dis)order and activity in the Discussion section (page 15-16)

Although the authors are now in a good position to describe the tertiary structure/inter-subunit interface at the PSM in a more biologically relevant setting (in contrast to crystal structures of fragments), I feel description in this area is scarce. It would be beneficial to describe this aspect, if possible (without going over the word/page limit specified by Nat. Comm.). Many crystal structures of all BphPs (with canonical domain organisation) have been determined as

fragments. Therefore the correlation observed in the past (Pr: bent/kinked overall PSM conformation, Pfr: more straight conformation) wasn't definitive (fragment construct/crystal packing). This new pair of DrBphP cryo-EM structures overcome that limitation. This is perhaps not unimportant since the change in dimer interface is what causes the breaking of the DHP interface.

We agree to the reviewer and have inserted a short discussion of this question on page 12.

Supplementary document, Fig. S2 and S4: On the particle selection and refinement process. There are 3 classes of maps generated each for Pr and Pfr following the ab initio reconstruction. It seems arbitrary to choose one class for both Pr and Pfr, which were then refined and used as a basis of discussion. But the %representation was not the criteria for choosing the class for subsequent refinement and discussion. E.g. For Pr, the class 2 represents higher proportion (41.8%) than that of class 1 (subsequently refined, 37.1%).

For Pfr: Ignore the class 1 (20.0%) since it represents Pr: class 3 (48.3%) represents higher proportion than class 2 (subsequently refined, 31.7%).

When protein conformation (and difference between Pr & Pfr) is under discussion, it appears strange to take minority conformation species for both Pr and Pfr as a basis of discussion, especially when classes show very different conformations (apparently much more so than for example a case of AtphyB by Li et al 2022). Shouldn't the variety of conformation be part of the discussion too?

We thank the reviewer for the important point. In response we have redrawn the two figures for increased clarity and we have included more information about the maps and different models. Briefly: In the dark we see two conformations (next to the highest resolution map of Pr), which have either much lower resolution or are missing the output domains. Under red light we see a model which resembles the Pr state (as it is expected since not all Pr particles will be converted), a class of "broken" proteins, and a class which we assigned to Pfr state based on the tongue conformation. The Pr and Pfr models were then refined in several steps to separate them from each other. Despite serious attempts, the Pr model from the red-light data never refined to better than 7Å. We have added this to the main text on p8 and 9 and to the figure captions of S2 and S4.

A related question to the above topic

Are there any possibility that some of the particles in the Pfr grid represented Pr/Pfr heterodimer?

Indeed, one would think that there should be some, but we did not identify any (from the tongue confirmation), despite not implying symmetry in the refinements. We did not impose symmetry in the refinement in order to capture mixed Pr/Pfr particles, but this was not the case. This is now discussed in a paragraph in the discussion section.

P5, Fig. 1 Legend: The effect of chimeric fusion on the phosphatase activity, ... because of competition between the fused and free DrRR.

There is no evidence that the mode of interaction between DrBphP and DrRR in the context of chimeric fusion dimer protein is the same as that of the inter-molecular interaction between two separate proteins. From the practical point of view, we may not need to understand the reason why the chimeric fusion strategy led to stabilization of the protein to render it suitable for cryo-EM. From the point of view of those wishing to learn from this manuscript, however, it will be useful to know whether it was indeed the biological interaction which was the key to stabilize the sample for cryo-EM. While not essential, it will be preferable if this question is addressed (although it will be difficult to discern between intra- and inter-molecular interaction).

We agree that the evidence supporting of native-like DrRR interaction was missing in the manuscript. We have now included a dark reversion experiment of DrBphP-DrRR fusion to the manuscript (Figure 1c). This experiment shows that the fusion protein has a faster dark reversion rate than the *DrBphP* by itself. The result is consistent with the findings of Multamäki et al. 2021, where the addition of free DrRR to a solution of DrBphP increases the thermal reversion of the phytochrome component. As fused DrRR induces a similar response to free DrRR, one can propose that the interaction between the proteins is similar. Fused DrRR has a higher impact on the dark reversion than free DrRR (Supplementary Figure 1b), which can be explained by high local concentration of the fusion.

In addition, the Pr-state cryo-EM density presented in this study hints for approximate places of the CA and DrRR domains. These places resemble the ones found, e.g., in the HK:RR complex crystal structures (PDB code 3DGE, see figure below), which is now mentioned in the text (page 7). This resemblance suggests that the interactions here may mimic the natural ones. We however note that our assignments of REC and CA remain tentative.

P10 line 14 and p12 line 18

How the value of 0.4 Å was derived isn't clear. (e.g. If two PHY domains (only) are superposed with one another and their centres of mass are compared, the distance between the centres will

naturally be very small) Was superposition carried out based on the entire PSM or only nPAS-GAF?

Thank you for pointing out this unclarity: We did not superimpose the domains at all, but measured the “distance between the center of mass between the PHY domains across the dimer”. Then we compared the distance for the Pr and Pfr structures and arrived at the difference of 0.4Å. This was not clearly stated in the first version, but is now corrected on page 10.

P3 Line 20

... in light-activated and resting states 7-9

Essen et al 2008, Yang et al 2008, and Wagner et al 2005 (2VEA, 3C2W, and 1ZTU) are all in resting states. (PaBphP is a bathy-phytochrome, Pr is its activated state, for both photoconversion or HK activity)

Noted and correct, changed to “Pr and Pfr states”

P4 line 5

... daft of structures ...

“Daft” sounds inappropriate, could you please check? Dearth? But even the latter isn’t so common.

Thank you, changed to “lack” . .

P16 line23 & p17 line 3, and Table S1: Initial model used

4Q0J (DrBphP in Pr) is consistently described as the initial model. Was this also true for the cryo-EM structural determination for DrBphP in Pfr? One imagines either 4O01 or 5C5K (DrBphP as illuminated/Pfr) are more suitable as initial models.

Thank you for pointing this out. For Pr we used 4Q0J, but for Pfr it was both 4Q0J and 5C5K as templates. This was because the density readily matched the overall domain arrangement of Pr-state structure 4Q0J, but the tongue and BV surroundings were much closer to 5C5K. We have clarified this in the text.

Figure S3 Leged: On the difference between cryo-EM and crystal structures (Pfr)

... but that the Pfr crystal structure does not (overlap)

Any two proteins can be superposed with one another, but the question is how good the superposition is. Would it possible to describe this by quantity? E.g. superposition using the entire PAS-GAF-PHY polypeptide (RMSD). In this case the value calculated for Pfr is expected to be larger than that of Pr.

Agreed, we have now quoted the RMSDs.

P12 line 8: it may be that several mechanisms of signal transduction into the effector domains coexist.

While this is a simple explanation, the picture becomes more complicated when one considers past studies involving a chimera based on Cph1 phytochrome (Levskeya et al 2005, <https://doi.org/10.1038/nature04405>). This chimeric enzyme (PSM of Cph1 and HK of EnvZ) exhibit light regulated enzyme activity, therefore the mechanism of signal transduction appears to be the same (or at least compatible) between Cph1 and EnvZ. EnvZ is a HK which features a HAMP domain, and HAMP domain was proposed to transmit signal via rotation (Hulko et al 2006, DOI 10.1016/j.cell.2006.06.058). This presents a strange picture where Cph1 (argued to operate via the neck opening in the present manuscript) sensory module is working with EnvZ (argued to work via the rotational model). Does this mean Cph1 can both break at the neck and rotate upon Pr/Pfr? Other similar studies utilising HK chimeras exist. This probably connects to modularity of HK domains and evolution of HK via domain swapping in prokaryotes. While one appreciates the difficulty in covering this topic extensively, the discussion will become more informative by expansion, if possible.

We believe that the underlying understanding is that HKs are versatile and probably support different ways to translate a structural signal into a biochemical one. However, it is also reasonable to think of phytochrome photosensory modules as “actuators” that can provide the signal in different ways (we believe that this is what the reviewer refers to). Thus, the EnvZ histidine kinase could be activated by rotation of HAMP domains, and it could also be activated by the zipper mechanism - to know what is happening one really needs to solve the structures of those chimera proteins.

We have revised the discussion accordingly.

Reviewer #2 (Remarks to the Author):

This paper describes the cryoEM structures of the full-length *Deinococcus radiodurans* bacteriophytochrome (DrBphP) in both the Pr and Pfr forms. This molecule has served as a key subject for studying many aspects of phytochrome light activation and signaling; understanding how it functions as a full-length protein, i.e. how the output signal is converted through its many domains to the enzymatic output is an important goal that will impact many fields, including biophysics, structural biology, signal transduction, photobiology and even plant biology, given the critical role of plant phytochromes in regulating growth and development. The key finding is that in the Pfr form, cofactor isomerization and structural rearrangement of the tongue element (which had been previously characterized) lead to an “unzipping” of the neck helices, that then causes increased conformational disorder and asymmetry (or dynamics) in the histidine kinase (CA) and receiver (REC) modules. Given that this is a full-length, multidomain signaling system, these results add substantially to what we understand about sensor histidine kinase regulation and complements the insights from previous studies. Unfortunately, the resolution and local quality of the Pfr structure is not sufficient to provide much more molecular detail on the structural changes that are propagated from the cofactor to the neck, as the Phy domain itself, which contacts the neck, changes minimally within the resolution of the structures. Overall, I do think this is an important work, but I do have some serious concerns, some technical, and some with interpretation, as described below:

Author response: We thank the reviewer for critically reviewing the manuscript and the comments provided. We have addressed those below.

We revisited the structure building and fitting and have thereby improved the quality of the model notably. We thank the reviewer for requesting this. With this we feel that the structures are much more solid now and hope that the reviewer agrees to this. The updated validation and coordinate file are available and attached to the submission.

1. Reconstructions: The quality of the density in the Pfr form, particularly in the neck, and also in the CA/receiver regions appears skewed and streaked, a possible consequence of orientational bias or even mixed classes. Orientational bias may indeed be indicated suggested by the FSC curves and the unusual bump around 5 Å resolution. The orientational sampling data for the structures is not provided, and should be.

We thank the reviewer for this suggestion. The data on orientational preference is now provided as Supplementary Figures 6 and 7. There are some orientational preferences, but overall the coverage of all angles is given.

Whereas there clearly seems to be increased disorder in the Pfr state, there is some question as to whether the unzipping of the helices can be solidly interpreted.

We see the concerns raised by the reviewer. We have carefully re-refined all the structures and updated the figures accordingly. We have also extended Figure 4 to visualize more clearly how the density supports the separation of the neck helices. The density encloses all modelled residues. We believe that the updated refinement and the new representation makes our interpretation of helix separation more convincing.

It's also unclear as to whether the structure is truly more asymmetric than the Pr state, or just more disordered. Only a preliminary validation report is provided for the structures. Are there problems that prevented submission of the finalized report? In that regard, the stereochemical statistics of the two models are somewhat worrisome, especially the percentile of the clashscore and sidechain outliers.

We thank the reviewer for the comment. We agree that the stereochemical statistics of the models were not ideal. We have therefore further refined the structures by applying higher structural restraints. This resulted in better sidechain outlier (<3.5% of poor rotamers) and clashscore (1.6-1.8) statistics. The pdb submission has passed the validation report and is ready for release.

The main argument for conformational signaling is heavily based on the alternative conformations of the zipper-like neck, particularly the distance changes of N513, L516, and N520 which locate on the dimer interface (figure 4); however, the validation reports suggested in both models, that those residues have unfavorable conformations.

The newly refined structures have now improved structural quality. The neck part (involving residues 513, 156, and 520) has less structural outliers. Although absent in Pr structures, some of the neck residues in the positions 513-520 still have unfavorable conformations in Pfr state, which can be explained by the disorder and incomplete density of the region.

Indeed, the absolute conformation of the side chains in the neck should be considered with care as the electron density is ambiguous and of low resolution in some parts of the structure. Therefore, we have removed the results about the change of hydrogen bonding at around N513 in the text. However, the main chain/backbone positions can be interpreted with good precision and also the opening of the helices is robustly encoded in the data.

Also, the zipper region does not fit very well to the density, especially in the Pfr state (e.g. fig. 4b bottom panel, the left alpha-helix looks displaced), which may be caused by the moderate quality of the modeling, again implied by the clashscore. It is convincing that the two models have significantly different conformations that propagate throughout their dimensions, but the quality of the models could be improved so as to avoid unnecessary bias in the interpretation.

We have now improved the models further and updated the figures containing the final deposited structures. The newly refined structures fit the neck/zipper density much better, which is also visible in the updated figures (Fig 4).

2. The biochemical experiments of Fig. 1c are somewhat difficult to interpret, more description in the legend would help (for example, it is not stated that the phosphorylated p-DrRR is added exogenously in the legend). Why is there substantial phosphatase activity of the DrBphP in the D state, nearly as much as in R?

We thank the reviewer for the comment and agree that the biochemical (ProsTag) experiment can be confusing when not explained carefully. We have therefore included a more pedagogic description in the legend. The unfused DrBphP show notable dark-state background activity in our hands, and the effect is even more pronounced if longer incubation time is used. We chose to present a gel with relatively long incubation time in Figure 1c because this shows increase in activity in the Pfr-state of DrBphP-DrRR more clearly. We have now included a full gel in Supplementary Figure 1c, which includes a shorter incubation time for transparency.

3. The paper sets the context by stating the importance of sensor kinase signaling, and in several places refers to other conformational mechanisms of sensor kinases in comparison. However, DrBphP has no kinase activity, it is a light-activated phosphatase and hence this point should be made in considering any comparison to other histidine kinases. There is the general question of what role the CA domains play here at all and whether a change in their order is relevant to phosphatase activity. It is true that for EnZ the CA domains enhance phosphatase activity of the Dhp domains, is this known for Dr Bphp as well? Overall, the paper should address these mechanistic issues more clearly and in particular how the structural changes observed relate to the key issues of this specific protein's activity before generalizing to the entire family of sensor kinases.

We agree to the reviewer that one cannot generalize the structural mechanism identified for DrBphP to all histidine kinases/phosphatases. We have revised the key section in the discussion and at a few other places in the document. We now even more clearly state that HK and phytochromes probably support several mechanisms of action (depending on the needs of the organism and the structure of the protein). Phytochromes are successfully used in chimera proteins (with HK and other output domains) and HKs generally are diverse, widespread and have evolved through domain swapping. There ought to be some versatility in the structural mechanism, and we believe that we convey this better now.

The CA domains with bound nucleotide are required for phosphatase activity (Multamäki et al. 2021).

Line 194 - The work of Reference 26 is an interesting comparison, but more discussion is warranted so as not to be misleading. Although the inhibited state of LuxQ is proposed to be asymmetric, the observed asymmetry occurs in the periplasmic sensing domain. The sensing domain of BpHp, appears to roughly symmetric, as least not nearly as asymmetric as LuxQ, so

its unlikely an asymmetric signal on the order of LuxQ could be involved here. Nevertheless, the Pfr structure may provide some evidence for an asymmetric cytoplasmic domain conformation in the phosphatase form, which would indeed be relevant to LuxQ.

We acknowledge the reviewer for pointing this out. Since the Neiditsch paper does not solve the structural change in the HK domain (and not even in the membrane-crossing helical bundle), it is not clear if the HK domains lose symmetry. It is therefore difficult to directly compare our structures to that paper. The recent work by Gardner (our citation 27) however shows that the helices and interactions with CA domains become destabilized in the active state (in a monomeric HK), which agrees quite well with our observations. We have adjusted the introduction about this accordingly. (page 3)

Lines 179-180 – more discussion is needed here – given the sentence before, one might conclude that this study contradicts the findings of Reference 34. However, the molecule studied in 34 is a diguanylate cyclase; the output domain is very different than in the phytochrome in that the enzymatic dGC is an obligate dimer and hence this oligomerization will likely influence any conformational changes in the connecting helices. It is a very interesting comparison, but these mechanisms need not be in opposition – they almost have to be different.

We thank the reviewer for pointing out this section, which could be misunderstood. We have changed the text so that it becomes more specific and to clarify what we mean with the sentence.

Lines 186-190 – with respect to scissoring, it seems to me that density in Pfr compared to Pr could be interpreted as a scissoring motion, or at least have a component of a scissoring motion, I also don't think that helical rotations can be fully ruled out at this resolution, especially given the concerns with the modeling and stereochemistry of these regions, as stated above.

We agree; it is not ruled out that other mechanisms co-exists and this is also reflected even more in the new version of the paper. The scissoring motion has been mainly suggested by the structures of Gordeliy on the transmembrane helical bundle of a HK; the changes are overall much smaller than what we observe in the present case. Thus, the opening motion that we observe could be seen as an addition to previous proposal.

Lines 193-194 – it's clear that order is lost, but less clear that symmetry is also lost, or rather that asymmetry is really a key aspect of the activation mechanism. The density in the validation reports looks reasonably symmetric, perhaps less so than with Pfr, but with a broader conformational ensemble, classification is going to be more difficult (I note that a lot of particles are excluded in the final reconstruction) and the result of a slightly asymmetric looking particle more likely.

We thank the reviewer for this comment. We have revised the manuscript to make sure that "order" and "symmetry" are used accurately. Symmetry is an important issue for understanding the structural mechanism. Flexible output domains are going to be asymmetric by definition,

thus we would prefer to keep the conclusion that “symmetry and order” is lost. Whether the loss of symmetry is implied in the phosphatase function is another question, which cannot be easily answered by our data.

Lines 58-62 and later 196-202 – I do think it is a significant finding that there is little change in the phy domains to propagate the signal, but the discussion with respect to different types of allostery could use some more development. Lines 58-62 – both “a cascade of structural changes leading to signaling” and “allosteric coupling between modules” fall under the modern explanation of allostery, which really comes down to coupled conformational changes at separated (binding) sites. There can be many mechanisms, involving a few specific moieties, or a more global “refolding” of domains and these altered interactions can certainly couple across interfaces, but all of these mechanisms can be considered allostery – perhaps more specifics need to be provided to allow the reader to understand better what is being distinguished.

We thank the reviewer for this comment - we have rewritten the paragraph in line with what the reviewer suggested/criticized and to stay closer to the new structures obtained. We believe that we agree on this issue and that the paper has improved by the revision.

Line 184 and Fig. S3 – *A. thaliana* PhyA is more closely related to *At* PhyB than DrBphP, and the recent cryoEM structure of PhyB shows the neck region to be in a completely different conformation than it is here – it would seem that a homology model for PhyA based on PhyB is more appropriate.

We have now included an AlphaFold model of *At*PhyA in Fig. S3, which is essentially identical to a homology model based on *At*PhyB. We have also changed the discussion about the applicability of the neck opening mechanism to plant phytochromes to acknowledge that there may be other mechanisms at play.

Last sentence – “inspire the development of new antibiotics and new optogenetic tools” – it would be helpful to say a little more about these points. There’s really no discussion of pathogenesis in the paper or why phytochromes would be a good target for drugs and how these structures would help with that, likewise for the optogenetic tools.

We thank the review for the comment and have removed the reference to antibiotics from the paper. The connection to optogenetics is obvious in our mind: the engineered phytochromes have already been used as fluorescent labels in mammalian cells (Shu Science 2009) and in optogenetic applications (Gasser et al., PNAS 2014; Gomelski et al., PNAS 2014) and we believe that the new structural knowledge in this paper will inspire further efforts in this direction. We have added the mentioned publications in PNAS to the last sentence to emphasize this.

Data processing – does the class for the Pr state extracted from the red-light-treated ensemble, match that of the pure Pr state?

The overall folds of the “Pr” model under red light agree with the model from the dark data. Also, its electron density in the tongue region indicates a beta sheet form. However, the resolution of the “Pr” state under red light was only about 7Å and we have therefore not pursued refinement of the model.

Figure S5 is quite confusing, and the assignment of the CA and REC not very convincing – was there any computational docking attempted to better validate the placements? Or perhaps attempts at a focused refinement of the map on the area where the domains reside?

We thank the reviewer for this comment and have redrawn Figure S5 for clarity. The electron densities were assigned based on the published HK:RR complex crystal structures and the difference sizes of the CA and REC domains (this is especially clear for the Pr state; panel a of the new figure). One of the REC domains appears missing in the density for the Pfr states, which we ascribe to the influence of disorder on the refinement process. Focussed refinement of the output domains alone did not result in improved electron densities.

Minor:

Abstract “which sensing is archived” is an unfamiliar formulation, consider rewording

Thank you, revised.

Line 50 : perhaps ...“used as” near-infrared fluorescent markers...

Thank you again, revised.

Fig.2 – it is unclear what the arrow for the CA domain is pointing at – can the CA and REC domains be distinguished at all? Perhaps brackets would help? I’d also recommend using a different color for the model and the density so that they can be better distinguished – same comment for Fig. 3.

We thank the reviewer for pointing this out and we have redrawn the labels in figure 2 and 3; We have changed the color of the models and have also added a cross reference to figure S5 for assignment to the CA domains.

Lines 116-117 – would do you mean by this statement, that the helices of the Dhp could be connected in a different handedness? More explanation is needed.

We have added a sentence explaining this just before the relevant section. The Dhp domain consist of two helices (short and long), which are connected by a loop. Since the Dhp helices form the dimer interface, there are two possibilities to connect the short and long helices. This question has been studied for histidine kinase. Likely, DrBphB follows the Maritima HK, but

there is a small chance that the connection is actually wrong. We believe that being clear about this in the paper is the best alternative.

Line 192 – do you mean that multiple mechanisms coexist in for the same protein, or different proteins have different mechanisms?

We have rewritten this section in response to reviewer comments and hope that this has been clarified now. It should be that different mechanism exist for different proteins.

Fig. 3 – label residue 521 in the models. Part c, given the map at the BV is fairly low resolution a superposition with the 15Z conformation would help convince the reader that the conformation is different.

We have added the requested figures as figure S5c (for Pr density) and S5d (for Pfr density). Great suggestion!

Line 171 – “dramatical” should be “dramatic”

Thank you for proof-reading!

Line 180 – It doesn't seem like the residues on the interior of the helices are still buried in the model? – 516-520 move far enough away to accommodate water, and after that the helices are undefined.

We had difficulties understanding the comment, but assume that the question is related to our statement that “the proposed change-of-register is not so important for the zipper model”. In any case, we have rewritten that statement in reply to other comments.

Note that there are a few plural/singular issues and other typos throughout.

We have carefully read and revised the language of the paper, and hope that it has improved.

Reviewers' Comments:

Reviewer #1:

Remarks to the Author:

In this revised version of the manuscript, all the points and questions raised in the first version seem to have been addressed.

I have forgotten to mention that some response by plant phytochromes must be completely independent of the neck region as the truncated construct with only 1-450 residues of phyB is able to inhibit the hypocotyl elongation (Matsushita et al 2003, and Oka et al 2004). It is true however that under the normal circumstances plant phytochromes work as full-length proteins, and C-terminus of phyB have been shown to possess a discernible function (Qiu et al 2017). Therefore I do not feel the need for the text in the discussion to be modified in this topic.

Regarding a question on the criteria of choosing a class for refinement in the first version of the manuscript:

When the loss of symmetry is being discussed as being important for signalling, and if that loss of symmetry is concluded by a domain no longer visible in the density map of Pfr – then the invisibility of the output domain in class 2 of Pr should also attract attention, especially if class 2 is more representative than class 1 – so I thought. Proteins are flexible, and I'd thought cryo-EM is a better suited way than X-ray crystallography to capture ensembles of structures.

Authors attribute impurities and broken particles as the cause of incomplete molecules which were not further considered. In that case, I consider my questions answered.

Reviewer #2:

Remarks to the Author:

The improvement to the structure refinement resolve my most substantial concerns and the authors have done a good job addressing my other comments and those raised by the other reviewers.